# The human insula processes both modality-independent and pain-selective learning signals

**Björn Horing** *, Christian Büchel

Department of Systems Neuroscience, University Medical Center Hamburg-Eppendorf, Hamburg, Germany

* b.horing@uke.de

**Data Availability Statement:** All data files underlying the summary data are available from the Open Science Foundation at https://osf.io/7jbv3/, DOI 10.17605/OSF.IO/7JBV3.

## Abstract

Prediction errors (PEs) are generated when there are differences between an expected and an actual event or sensory input. The insula is a key brain region involved in pain processing, and studies have shown that the insula encodes the magnitude of an unexpected outcome (unsigned PEs). In addition to signaling this general magnitude information, PEs can give specific information on the direction of this deviation—i.e., whether an event is better or worse than expected. It is unclear whether the unsigned PE responses in the insula are selective for pain or reflective of a more general processing of aversive events irrespective of modality. It is also unknown whether the insula can process signed PEs at all. Understanding these specific mechanisms has implications for understanding how pain is processed in the brain in both health and in chronic pain conditions. In this study, 47 participants learned associations between 2 conditioned stimuli (CS) with 4 unconditioned stimuli (US; painful heat or loud sound, of one low and one high intensity each) while undergoing functional magnetic resonance imaging (fMRI) and skin conductance response (SCR) measurements. We demonstrate that activation in the anterior insula correlated with unsigned intensity PEs, irrespective of modality, indicating an unspecific aversive surprise signal. Conversely, signed intensity PE signals were modality specific, with signed PEs following pain but not sound located in the dorsal posterior insula, an area implicated in pain intensity processing. Previous studies have identified abnormal insula function and abnormal learning as potential causes of pain chronification. Our findings link these results and suggest that a misrepresentation of learning relevant PEs in the insular cortex may serve as an underlying factor in chronic pain.

## Introduction

Apart from its role in signaling tissue damage, pain is increasingly considered to be a preeminent teaching signal [1,2] in the context of reinforcement learning models [3]. For example, delta rule learning models in classical fear conditioning, such as the Rescorla–Wagner model [4], almost exclusively employ pain as unconditioned stimulus (US). In this and similar models, the value of predictive cues (conditioned stimuli, CS) is updated by the difference between the expected and the experienced outcome, i.e., a prediction error (PE). In this case, the PE

**Funding:** This research was supported by the German Research Foundation (DFG) grant SFB 289 Project A02 (Project-ID 422744262–TRR 289) (CB) and the European Research Council grant ERC-AdG-883892-PainPersist (CB). The funders had no role in study design, data collection and analysis, decision to publish, or preparation of the manuscript.

**Competing interests:** The authors have declared that no competing interests exist.

**Abbreviations:** CS, conditioned stimulus; EPI, echo-planar imaging; fMRI, functional magnetic resonance imaging; MNI, Montreal Neurological Institute; PE, prediction error; SCR, skin conductance response; US, unconditioned stimulus; VAS, visual analogue scale.

needs to be signed and signals the direction of the difference between expectation and event, i.e., whether the outcome is better or worse than expected. In the case of an aversive event like painful stimulation, this is relevant for shaping future behavior. Reinforcement learning particularly relies on these valences, and different neuronal correlates have been reported for aversive compared to appetitive PEs [5–8]. This has important clinical implications, as pathological learning mechanisms [1,9] have been reported in chronic pain.

However, PEs can also be computed as unsigned [10–12]. An unsigned PE simply indicates the presence of an unexpected event regardless of its valence. Unsigned PEs are therefore conceptually related to constructs like surprise or salience and may contain information concerning the urgency of behavioral change [13]. Computational models of learning can include either type of PE or both [4,10,14–16]—for example, the Pearce–Hall model incorporates the unsigned PE as a factor to increase the learning rate after highly incongruent (surprising) events [14,17], whereas a hybrid model contains both terms [10,17,18].

Previous studies investigating PEs in the context of aversive learning have observed signal changes in the anterior insula related to unsigned PEs [6,12,19–21]. Unfortunately, in many studies, a signed PE signal is nonorthogonal to stimulus expectation, which poses a problem with a short interval between CS and US, and the low temporal resolution of functional magnetic resonance imaging (fMRI). Consequently, these studies were suboptimal to investigate signed PEs.

Granted that unsigned PEs resemble a surprise signal, they could plausibly involve similar regions for all surprising events, independent of the stimulus sensory modality. Crucially, the representation of unsigned pain PEs in the anterior insula [12,19] raises the question of whether these are specific to pain or simply related to aversive events. Control conditions using comparator modalities are essential to tease out the unique contributions of painful stimulation to the observed cerebral activity [19,22–24]. In such designs, modality PEs can be an important source of variance. Another critical question is therefore to understand which brain regions are active during the processing of such modality PEs, which is an understudied aspect given the relative scarcity of cross-modal experiments.

To further investigate the existence of signed PEs and the modality specificity of unsigned PEs, as well as the underlying neuronal mechanisms, we used a Pavlovian transreinforcer reversal learning paradigm [25,26]. This involves 2 visual stimuli as CS and 2 intensities of painful heat or loud sounds as US (for brevity, these are referred to as "pain" and "sound" forthwith). Across sensory modalities, stimuli were chosen to be roughly comparable in salience as indicated by similar skin conductance responses (SCRs) [27]. Reversals occurred between US intensity but within US modality (e.g., CS predicting low pain will next predict high pain) or within US intensity but between US modality (e.g., CS predicting loud sound will next predict high pain). Analyses focused on PEs within and across modalities, using advanced surface-based analyses of high-resolution fMRI together with SCRs.

We expected that SCR resembles unsigned PEs, as SCR is generally considered to reflect arousal-related activation [27–29] and thus the sign of the PE—representing its valence—should not affect it. Concerning fMRI and following results from previous studies using painful stimulation to investigate PEs, particularly in a multimodal context [12,19,22,24,30], we focus on a region of interest including insular and opercular cortices contralateral to stimulation, while also reporting whole brain results. For the processing of pain, the insula is of particular importance given its rich structural and functional (somato)sensory connections, including strong internal connectivity [30–33]: The dorsal posterior insula has been demonstrated to have a preferential involvement in painful stimulation [22,34–37]. However, insular processing especially in anterior segments also occurs across sensory modalities, i.e., has been implicated in multimodal integration and the processing of supramodal dimensions like

unpleasantness, salience, and PE processing [23,38,39]. Concerning PEs, we expected to replicate previous results [12,19] showing the representation of unsigned PEs in the anterior insula. More importantly, we expected that this signal occurs independent of the modality of the US (i.e., both for sound and pain). In agreement with this nonspecific response, we also expected modality PEs to be represented in the anterior insula. However, in this case, the magnitude of the difference could feasibly be weaker or stronger than the intensity PEs: A weaker signal might arise as the intensity—and thus salience and other general aspects—are intendedly not different between the expected and the received US; a stronger signal could arise if the qualitative change between the 2 modalities dominated the cerebral responses.

Because signed intensity PEs have a direct conceptual overlap to systems like reward (or relief) and punishment, as well as a large implications for adaptive behavior, we have placed another focus on their cerebral correlates. Employing our novel paradigm, we were also in the position to investigate signed intensity PEs. Focusing on pain, we expected them to be either represented as a distinct part of the anterior insula or within the mid to posterior insula. The former is suggested by inherent differences in salience between the 2 intensities and the latter by the notion that a signed PE necessitates some form of intensity encoding, which has been observed in the dorsal posterior insula [22,35,40,41].

## Results

In 2 sessions with 64 trials each, 47 participants learned associations of 2 CS (fractal pictures) with individually calibrated US (2 painful heat intensities and 2 loud sound intensities) (Fig 1A and 1B). In each trial, either CS appeared, followed by symbols of all 4 US, from which participants selected the US they expected (Fig 1C). One of the US was then applied. CS–US associations were deterministic, but importantly, associations frequently reversed and had to be relearned over the course of the experiment (Fig 2). Reversals occurred unannounced after a

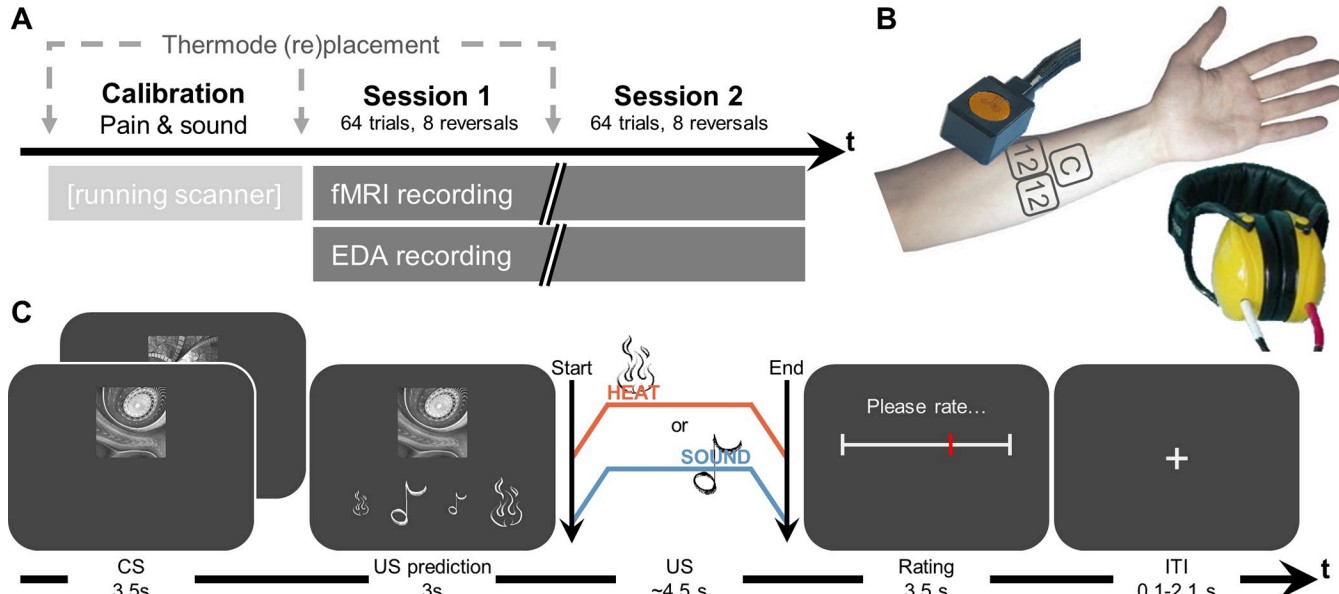

**Fig 1. Experimental protocol. (A)** Overall structure of the experiment. Calibration took approximately 15 minutes, each session approximately 20 minutes. **(B)** Devices used for heat stimulation (thermode) and sound stimulation (headphones), with standardized locations on the left arm for pain calibration and either of the 2 experimental sessions. **(C)** Trial structure with associated durations. After displaying CS, participants were asked to choose which US they expected to follow. The US was then applied and rated in terms of its painfulness (for pain)/unpleasantness (for sound). EDA, electrodermal activity; CS, conditioned stimuli; US, unconditioned stimuli.

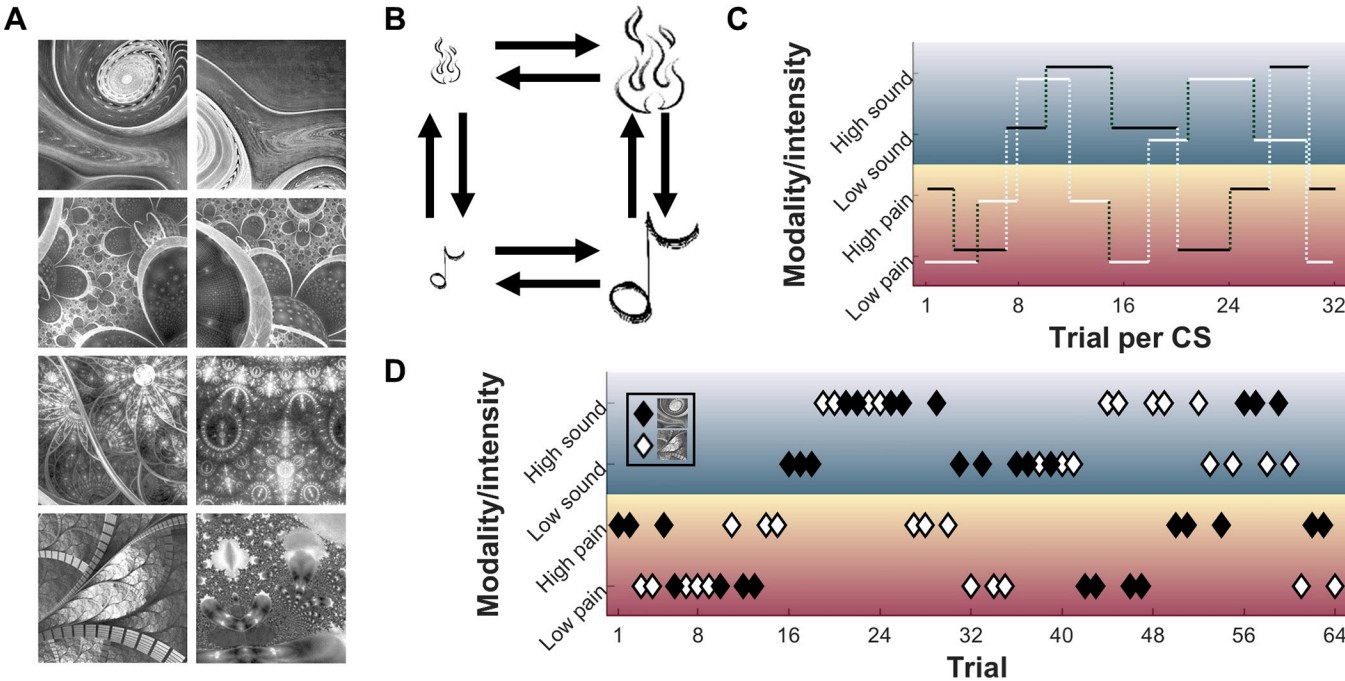

**Fig 2. Learning protocol aspects of the experiment. (A)** Set of CS; 2 were randomly selected for each participant (constraint: stimuli in row 2 could never both be selected due to high similarity). **(B)** Possible US associated with a CS at any particular trial (low pain, high pain, low sound, and high sound). Arrows indicate possible reversals; notably, no combined intensity and modality (cross)reversals occurred. **(C)** Example for contingencies of CS1 (black solid line) and CS2 (white solid line) for their 32 trials per session each. Vertical dotted lines indicate reversals, with light dotted lines for modality reversals and dark dotted lines for intensity reversals. **(D)** Example for an actual trial sequence of 64 trials with interspersed CS1 (black diamonds) and CS2 (white diamonds) and their associated US (rows). Data used to produce the figure can be found at https://www.doi.org/10.17605/OSF.IO/7JBV3. CS, conditioned stimuli; US, unconditioned stimuli.

randomized number of trials. Reversals could occur along the modality dimension or the intensity dimension, but not both simultaneously (e.g., no low heat to high sound reversals). See Materials and methods and S1 Fig for further details concerning design and protocol.

### Behavioral results: Calibrated stimulus intensities

Calibration yielded temperatures of 44.4 ± 1.2°C for the less painful stimulus (25 visual analogue scale [VAS]) and 46.8 ± 1.2°C for the more painful stimulus (75 VAS). For sound, calibration yielded 91.7 ± 2.8 dBA for the less loud sound (25 VAS) and 97.9 ± 3.7 dBA for the louder sound (75 VAS). Distributions of calibrated stimulus intensities are displayed in S2A Fig.

### Behavioral results: Stimulus ratings

The first question concerning the behavioral data was whether ratings corresponded to the calibrated intensities (supposed to yield VAS of 25 and 75, respectively). Actual low pain ratings were at 15.4 ± 14.8 VAS and high pain ratings at 66.8 ± 21.3 VAS; low sound ratings were at 29.2 ± 21.0 VAS and high sound ratings at 63.3 ± 19.4 VAS (Fig 3A; see S2B Fig for individual ratings per participant).

### Behavioral results: Learning performance

The next behavioral question was whether the participants learned the CS–US contingencies. Fig 3B depicts mean performance in predicting the US currently associated with the CS, in relation to the reversals of the association. Combining reversal types and comparing performance

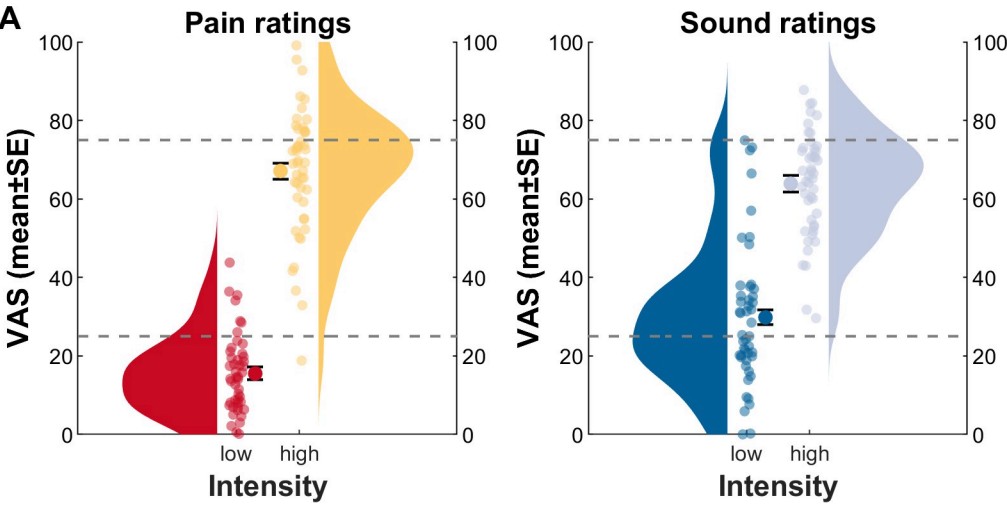

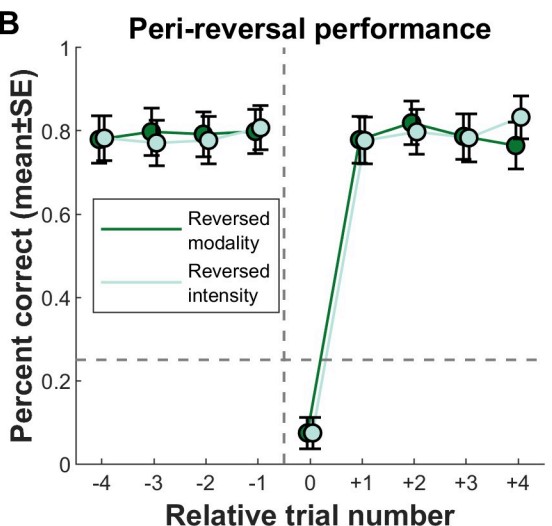

**Fig 3. Behavioral results for pain ratings and performance. (A)** Results for low and high unconditioned pain and sound stimuli; aggregate ratings of all pain and sound trials. Circles with error bars show the mean ± standard errors over all participant means. Participant means are displayed as smaller circles. Violin plots aggregate over participant means. The gray dashed line is the "intended" rating as per calibration (VAS 25 for low and VAS 75 for high intensities). **(B)** Performance pre- and postreversals, aggregated over all participants. Circles indicate the performance during (peri)reversal trials, first averaged within and then between participants (mean ± standard errors). The dashed horizontal line marks chance level (25%, i.e., 1 of 4 options). The dashed vertical line indicates contingency reversal, with relative trial number 0 as the reversal trial. Note that no difference arose between trials preceding and following modality versus intensity reversals (also see Fig 2 for aspects concerning contingency reversals). Furthermore, the steep increase in performance after trial number 0 indicates, on average, rapid learning of the new contingency. Data used to produce the figure can be found at https://www.doi.org/10.17605/OSF.IO/7JBV3.

at the single trials prior to reversal, at reversal, and after reversal, we find prereversal performance to be above chance level (t[79] = 13.8, $p \approx 0$), at reversal performance below chance (t[79] = −15.9, $p \approx 0$), and postreversal performance back above chance (t[79] = 19.5, $p \approx 0$).

Using a linear mixed effects model with reversal type (modality versus intensity reversals) and trial as categorical predictors, we find no mean difference of reversal type ($p = 0.94$) but of trial ($p = 1.2 \times 10^{-7}$). Post hoc contrasts indicate that the trial effect is driven exclusively by the reversal trial (compared to all other trials, all $p < 1 \times 10^{-169}$), whereas none of the nonreversal trials is different from each other (all $p > 0.1$).

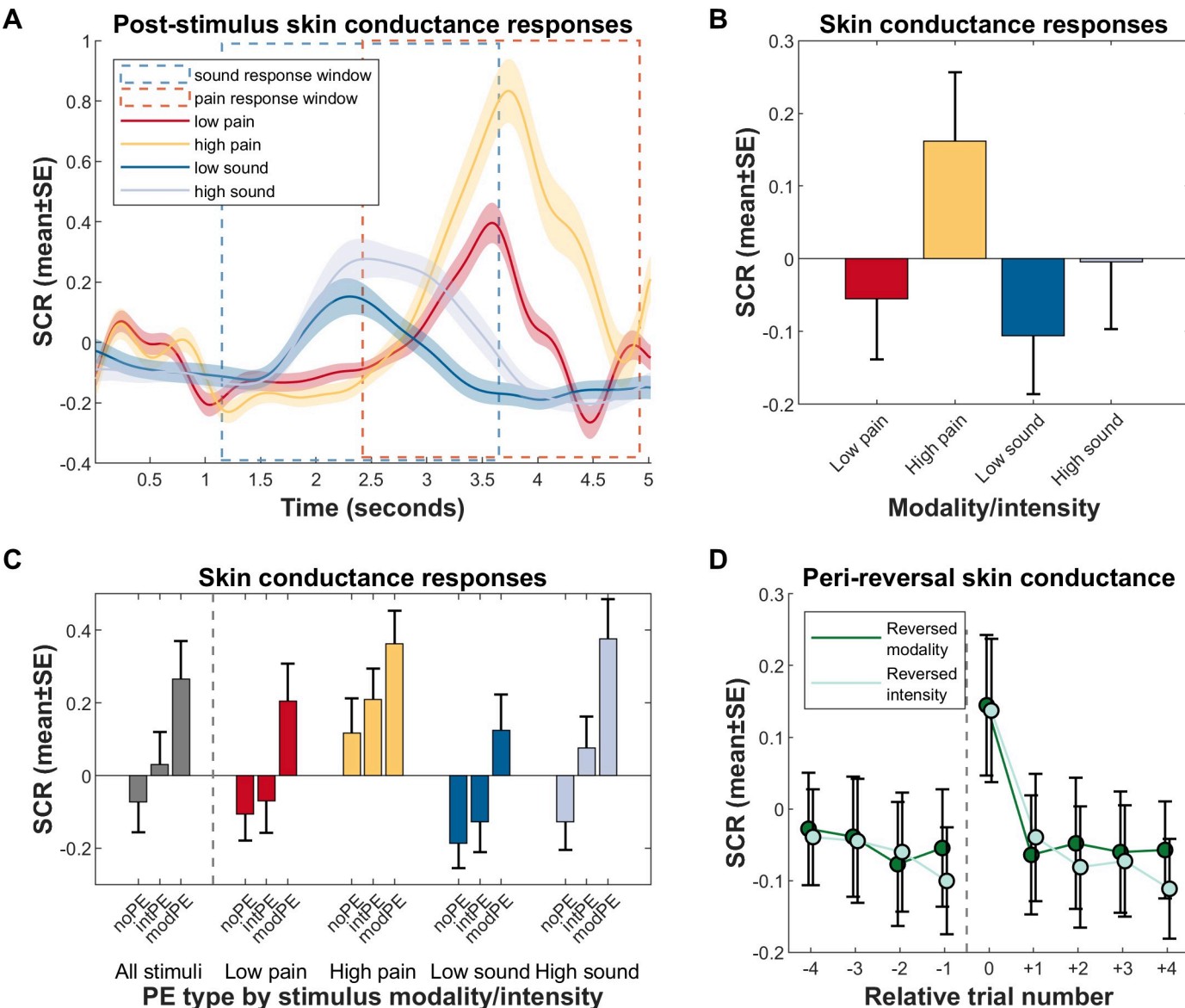

**Fig 4. Results from SCR measurements.** All plots are based on log- and z-transformed data. **(A)** SCR in relation to US onsets, by US modality/intensity. Note the differences in latencies between the 2 modalities (pain in red/yellow has a later onset, sound in dark blue/light blue earlier), which determined the response windows used for mean SCR calculation in panel b. **(B)** Mean SCR by US, calculated within each modality's response window. On average, SCR is not significantly different between modalities; differences arise between intensities, and in the interaction of modality and intensity (see text for parameters). **(C)** Mean SCR by US and PE type. Over all modalities and intensities, differences arise between each PE type. Within specific modality/intensity combinations, differences between no PEs and intensity PEs only arise in the high sound condition. **(D)** Mean SCR in and around reversal trials. Within trials, data are pooled over all modalities, intensities, and expectations, i.e., does not consider whether participants correctly predicted the subsequent stimulus. The dashed vertical line indicates contingency reversal, with relative trial number 0 as the reversal trial. SCR rises sharply after reversal, but quickly adapts postreversal to a stable level. Data used to produce the figure can be found at https://www.doi.org/10.17605/OSF.IO/7JBV3. intPE, intensity prediction error; modPE, modality prediction error; noPE, correct prediction; PE, prediction error; SCR, skin conductance response; US, unconditioned stimulus.

## Skin conductance response results

The major question concerning SCR results were whether any differences between the US arose and how the different PE types would be reflected in this psychophysiological measure of nonspecific characteristics or processes like arousal, salience, or surprise. SCR following sound has a faster onset than that following heat pain stimuli (Fig 4A; see Materials and methods

concerning the different response windows). The average amplitude of pain-related SCR was higher than the average of sound-related SCR, but this difference only showed a trend toward significance (main effect modality, t[4399] = −1.7228, $p$ = 0.08499; random intercept linear mixed model predicting each participant's and each trial's SCR). Instead, the difference is subsumed by a larger difference between low and high stimuli in the pain modality, as compared to that in the sound modality (modality*intensity, t[4399] = −2.9739, $p$ = 0. 0029567). On average, higher stimuli lead to larger amplitude as well (main effect intensity, t[4399] = 8.2743, $p$ = $1.7 \times 10^{-16}$). Investigating this difference only in correctly predicted trials shows a similar effect on SCR (modality, t[2674] = −1.4379, $p$ = 0.1506; intensity, t[2674] = 8.0081, $p$ = $2 \times 10^{-15}$; modality*intensity, t[2674] = −4.6669, $p$ = $3 \times 10^{-6}$) (S3 Fig, S1 Table).

Further investigating SCR differences following PEs, we first distinguished SCR when participants correctly predicted the US from trials when either an intensity PE or modality PE was made (Fig 4C). The following statistics include all trials—not just reversals—where an incorrect prediction was made. As shown in the first block (gray bars), over all US and controlling for modality and intensity, SCR following unsigned intensity PEs are larger than those following no PE (intPE > noPE, t[4397] = 4.336, $p$ = $2 \times 10^{-05}$), while SCR following modality PEs are even larger (modPE > noPE, t[4397] = 12.345, $p$ = $2 \times 10^{-34}$; modPE > intPE, t[4397] = 6.398, $p$ = $2 \times 10^{-10}$).

Notably, we performed an adjunct analysis on whether the direction of intensity PEs (i.e., signed intensity PEs) had an impact. We obtained mean SCR differences per participant between no PE and intensity PE trials for each modality and intensity separately, thereby accounting for higher intensity-related base SCRs; next, we contrasted these (now signed) PE-related differences between the low and high intensity. For pain, results indicate no effect (PE-related SCR difference for low pain mean ± SE 0.036 ± 0.052, for high pain 0.0922 ± 0.0622, paired $t$ test t[36] = −0.725, $p$ = 0.4731), while for sound, a more ambiguous yet nonsignificant result arose (PE-related SCR difference for low sound mean ± SE 0.060 ± 0.054, for high sound 0.199 ± 0.054, paired $t$ test t[35] = −1.931, $p$ = 0.0616).

In 4 consequent analyses, we investigated differences in SCR following PEs in all US separately, meaning that all intensity PEs are now signed. Results indicate that the intPE > noPE effect of the global analysis is driven by this contrast in the high sound US (light blue bars, t[1119] = 4.732, $p$ = $3 \times 10^{-6}$; random intercept linear mixed model); it does not reach significance following any other US. Conversely, modality PEs are followed by larger SCR in all US (all modPE > noPE $p$ < 0.001; smallest effect modPE > intPE t[1090] = 2.045, $p$ = 0.041079).

Fig 4D shows the average perireversal trial effect on SCR, over all US. It shows a large increase in SCR during both modality and intensity reversals; note that this analysis does not consider actual participant expectation, just the position related to the reversal trial. SCR is highest during the reversal trial and rapidly reaches a lower plateau even one trial later. Comparing the prereversal trial to immediate postreversal (trials −1 to +1), SCR is not significantly different if a modality reversal occurred ($p$ = 0.54704); this is also the case if an intensity reversal occurred ($p$ = 0.071164).

## Imaging results

We first obtained an overview of modality-related effects (Fig 5A and 5B) and rating-related effects (Fig 5B and 5C) of the US. All locations are reported using Montreal Neurological Institute (MNI) coordinates ($XYZ_{MNI}$). As expected, heat stimulation was followed by larger activation in widespread insular and opercular areas, with the highest peak in the dorsal posterior insula ($XYZ_{MNI}$ 35.5/−17.9/21.4, T = 12.2, p[corr.] ≈ 0). Activation following sound stimulation peaks in the superior temporal gyrus ($XYZ_{MNI}$ 65.9/−23.8/10.2, T = 25.7, p[corr. wb.] ≈

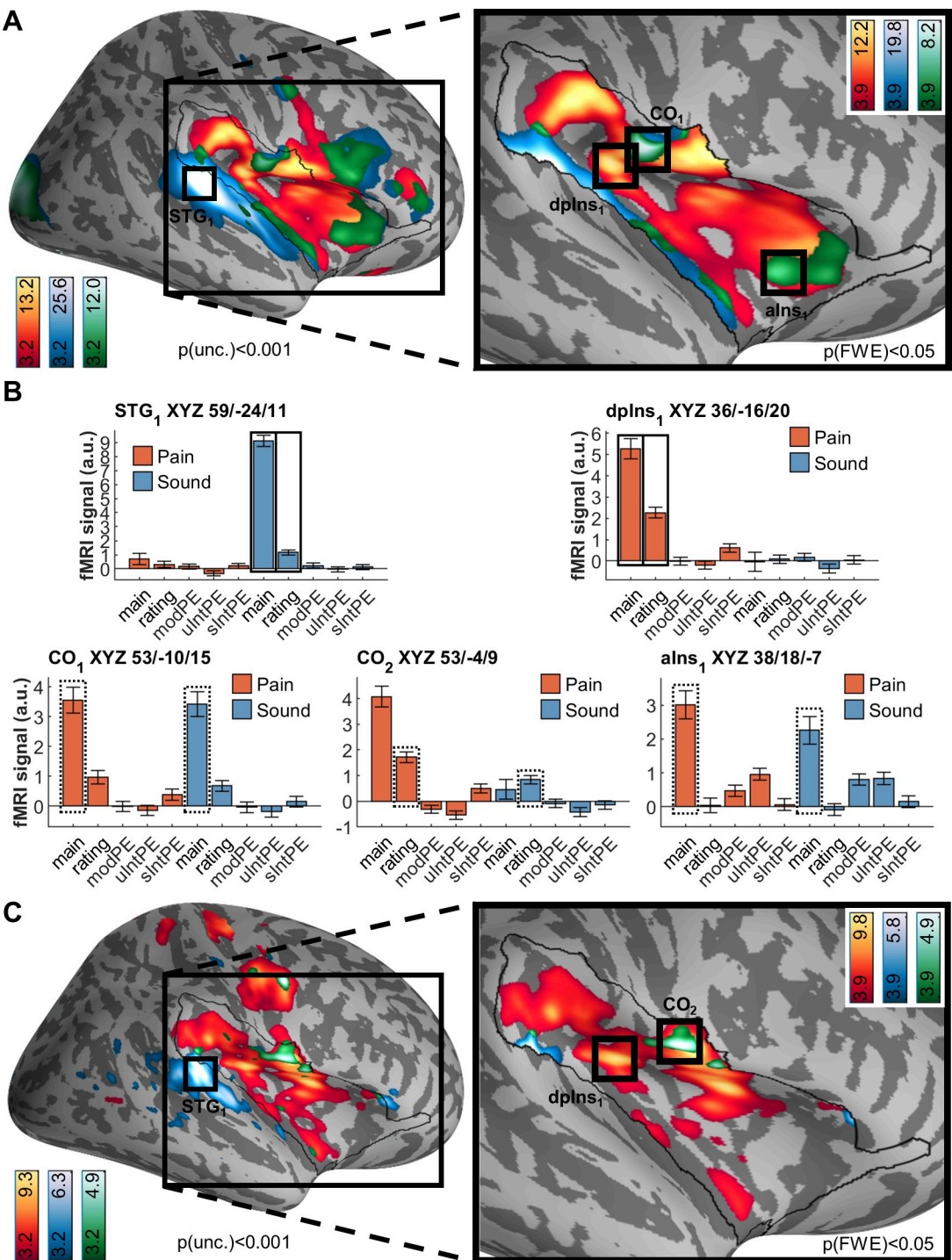

**Fig 5. Brain activation following pain (red/yellow) and sound (blue), including overlaps as per conjunction analyses (green).** Activations are overlaid on an average brain surface; for display purposes, activations in the whole brain lateral view are thresholded at p[uncorr.] < 0.001. The black line in the zoomed-in view delineates the region of interest and includes activations within the small volume FWE corrected at p[corr.] < 0.05. Peaks are shown for small volume only; bar plots show beta weights of BOLD activation obtained from a general linear model (see Materials and methods) from the respective peaks. See Supporting information for peak positions in whole brain (S4 and S6 Figs) and brain volume slices (S5 and S7 Figs). **(A)** Differential and shared activation following painful heat stimulation and loud sound stimulation. Peak activation following heat is located in (peri)insular areas contralateral to stimulation, namely the dorsal posterior insula (dpIns$_1$), and extending through the central and parietal opercula. Peak activation following sound is located in the superior temporal gyrus. Common activation

(green) is located in the central operculum ($CO_1$) and dorsal anterior insula ($aIns_1$), among other regions. **(B)** fMRI signal (arbitrary units) for peaks detected in panel A (US onset effects) or C (parametric modulation by ratings). Black rectangles highlight the regressors used for analysis; solid line indicates analysis with the respective individual regressor, and dashed line indicates conjunction analysis. fMRI signal labels refer to the regressors used for each modality: "main" for main effects of modality, "ratings" for behavioral ratings, "modPE" for modality PEs, "uIntPE" for unsigned intensity PEs, and "sIntPE" for signed intensity PEs. **(C)** Differential and shared correlations with pain ratings (for heat) and unpleasantness ratings (for sound). Activation correlated with pain ratings is focused on the dorsal posterior insula ($dpIns_1$). Activation correlated with sound ratings is focused on the superior temporal gyrus. Conjunction activation peaks in central operculum ($CO_2$) and precentral gyrus. Data used to produce the figure can be found at https://www.doi.org/10.17605/OSF.IO/7JBV3. BOLD, blood oxygenation level dependent; fMRI, functional magnetic resonance imaging; FWE, family-wise error; PE, prediction error; US, unconditioned stimulus.

0), just outside the extended insula mask. Notably, a conjunction of both heat and sound main effects shows activation in the central operculum ($XYZ_{MNI}$ 53.0/−10.3/15.1, T = 8.3, p[corr.] = $8 \times 10^{-13}$), dorsal anterior insula ($XYZ_{MNI}$ 37.6/18.4/−7.0, T = 5.6, p[corr.] = $2 \times 10^{-05}$), and several regions in between peaks for both modalities.

Next, we tested for fMRI responses correlated with stimulus perception, i.e., pain and sound VAS ratings (Fig 5B and 5C). For pain ratings, associations arose in the dorsal posterior insula ($XYZ_{MNI}$ = 35.2, y = −17.4, z = 18.6, T = 7.2, p[corr.] = $1 \times 10^{-09}$). For sound ratings, we observed a peak directly adjacent to the small surface ($XYZ_{MNI}$ 59.8, y = −33.9, z = 5.4, T = 4.8, p[corr.] = 0.016). Common activation between pain and sound ratings peaked in the central operculum ($XYZ_{MNI}$ 53.2, y = −2.7, z = 8.9, T = 4.8, p[corr.] = 0.001). Of note, the central operculum peak ($CO_2$ in Fig 5C) is located slightly anterior to that found for the modality (main effect) conjunction ($CO_1$ in Fig 5A) but shows barely any sound modality activation; conversely, peak aIns1 indicates that no rating effects are encoded here. See Supporting information for additional activations (S4 and S6 Figs).

## Unsigned intensity prediction errors

Having ascertained strictly stimulus-related effects, our next analysis included an investigation of unsigned intensity PEs within and between either modality (Fig 6). The guiding question here was whether any differences and commonalities between the modalities would emerge. Since we used the actual expectation queried from participants, "prediction error" here means that participants explicitly expected one intensity but received the other. Consequently, the unsigned PE implies some extent of surprise.

In both modalities, widespread activation was observed. However, conjunction analyses revealed that the majority of the observed activation actually overlapped between the modalities (green in Fig 6). The anterior insula constituted the dominant cluster of this overlap, with symmetric bilateral peaks ($XYZ_{MNI}$ = 34.6/23.5/−1.5, T = 5.8, p[corr. wb.] = $1 \times 10^{-04}$); whole brain significant frontal (medial and lateral), temporal, and parietal activation was also observed (S8 Fig).

Two aspects were of particular interest to us considering unsigned intensity PE results: First, that brain activation related to unsigned intensity PEs (Fig 6) was distinct from the rating-related activation (Fig 5). Second, the fMRI signal of the common activation in the anterior insula clearly indicated that modality PEs are likewise encoded in this area.

## Modality prediction errors

Following these 2 observations, we proceeded to investigate the nature of the overlap between the 2 types of PE. Like with unsigned intensity PEs, we observed widespread activation following each modality PE separately (Fig 7). Likewise, all unimodal activation is subsumed in the conjunction analysis, which indicates a large dorsal anterior insula cluster in our region of

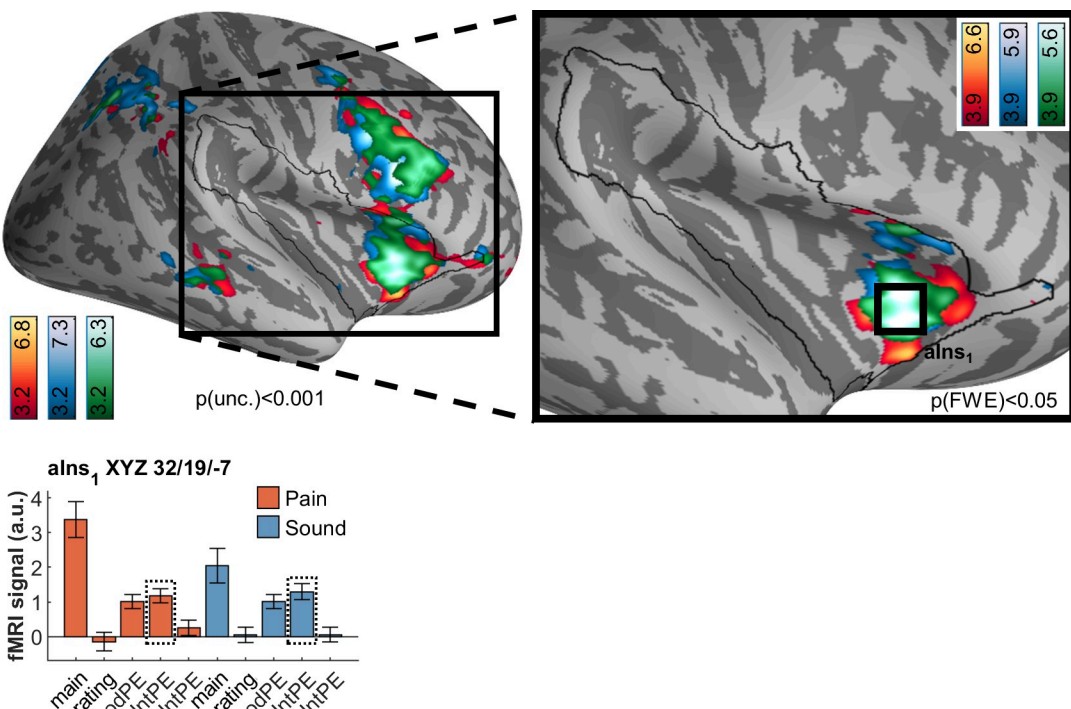

**Fig 6. Brain activation following unsigned intensity prediction errors in pain (red/yellow) and sound (blue), including overlaps as per conjunction analyses (green).** Peak activation following either modality is located in the anterior insula (aIns$_1$) and is subsumed in the common activation. Activations are overlaid on an average brain surface; for display purposes, activations in the whole brain lateral view are thresholded at p[uncorr.] < 0.001. The black line in the zoomed-in view delineates the region of interest and includes activations within the small volume FWE corrected at p[corr.] < 0.05. See Supporting information for peak positions in whole brain (S8 Fig) and brain volume slices (S9 Fig). In the fMRI signal bar graph, black rectangles highlight the regressors used for analysis; solid line indicates analysis with the respective individual regressor, and dashed line indicates conjunction analysis. fMRI signal labels refer to the regressors used for each modality: "main" for main effects of modality, "ratings" for behavioral ratings, "modPE" for modality PEs, "uIntPE" for unsigned intensity PEs, and "sIntPE" for signed intensity PEs. Data used to produce the figure can be found at https://www.doi.org/10.17605/OSF.IO/7JBV3. fMRI, functional magnetic resonance imaging; FWE, family-wise error; PE, prediction error.

interest (XYZ$_{MNI}$ 32.3/22.4/−3.4, T = 5.4, p[corr.] = $5 \times 10^{-05}$). Beyond this region, widespread common activation is observed, for example, in the superior parietal lobule, precuneus, temporoparietal junction, middle frontal gyrus and frontal operculum, and medial orbital gyrus (S10 Fig).

## Overlap of unsigned prediction errors

As a next step, we wanted to more formally assess the apparent overlap between both types of unsigned PEs. To do so, we simply computed the conjunction between unsigned intensity and modality PE (Fig 8). This analysis corroborated the anterior insula peak determined by separate analyses above. Furthermore, activation extended dorsally through the middle frontal gyrus and also included medial prefrontal areas adjacent to the dorsal anterior cingulate cortex.

## Signed intensity prediction errors

After ascertaining the effects for unsigned PEs for both intensity and modality, the final question for our fMRI data referred to differences and commonalities following signed intensity

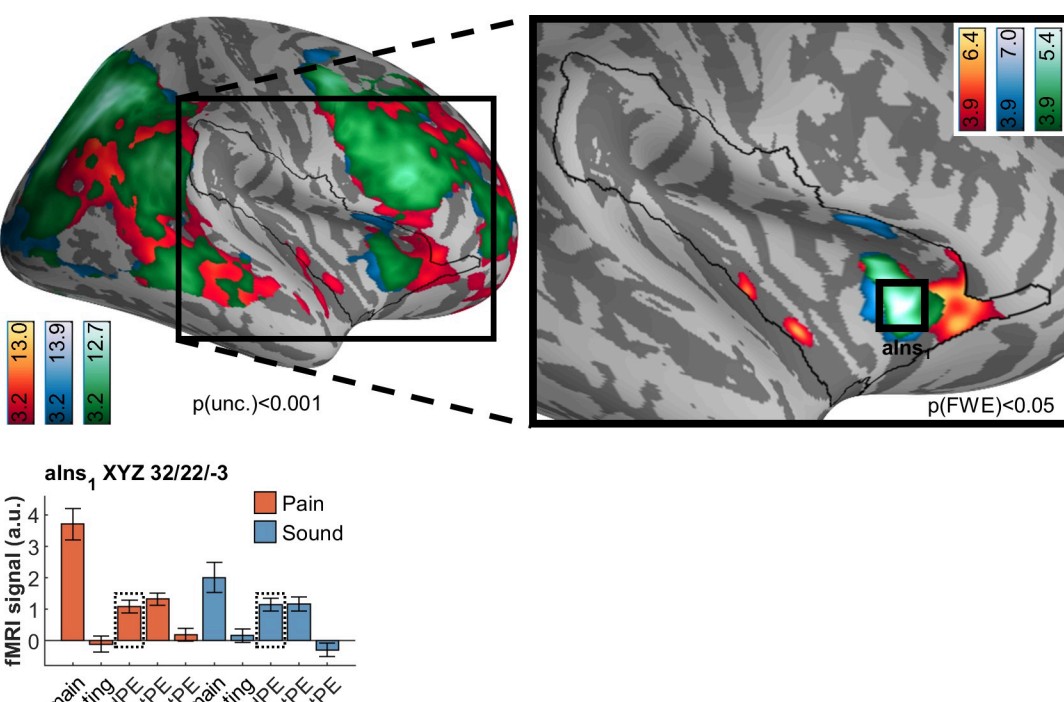

**Fig 7. Brain activation following modality prediction errors in pain (red/yellow) and sound (blue) activation, including overlaps as per conjunction analyses (green).** As with unsigned intensity PEs, peak activation following modality PEs in either modality is located in the anterior insula (aIns₁) and is largely subsumed in the common activation. Activations are overlaid on an average brain surface; for display purposes, activations in the whole brain lateral view are thresholded at p[uncorr.] < 0.001. The black line in the zoomed-in view delineates the region of interest and includes activations within the small volume FWE corrected at p[corr.] < 0.05. Peaks are shown for the small volume only. See Supporting information for peak positions in whole brain (S10 Fig) and brain volume slices (S11 Fig). In the fMRI signal bar graph, black rectangles highlight the regressors used for analysis; solid line indicates analysis with the respective individual regressor, and dashed line indicates conjunction analysis. fMRI signal labels refer to the regressors used for each modality: "main" for main effects of modality, "ratings" for behavioral ratings, "modPE" for modality PEs, "uIntPE" for unsigned intensity PEs, and "sIntPE" for signed intensity PEs. Data used to produce the figure can be found at https://www.doi.org/10.17605/OSF.IO/7JBV3. fMRI, functional magnetic resonance imaging; FWE, family-wise error; PE, prediction error.

PEs, i.e., correlations of brain activation with higher than expected intensity (Fig 9). For pain, we observed an activation in the dorsal posterior insula (XYZ_MNI 36.4/−17.3/15.8, T = 4.0, p [corr.] = 0.023). The dorsal posterior insula is an area considered of fundamental importance for the processing of pain intensity [22,35,42]. For sound itself, the peak activation was observed outside the region of interest, in the middle temporal gyrus (XYZ_MNI 49.4/−16.6/ −13.4, T = 4.1, p[uncorr.] = 2 × 10⁻⁰⁵) (see Fig 6). Within the region of interest, sound-related activation was found in the anterior insula (XYZ_MNI 36.7/11.0/−10.2, T = 4.2, p[corr.] = 0.015). Notably, these are adjacent to the unsigned PE activations (Figs 6–8). All signed intensity PE peaks, both for pain and sound, show no significant representation of a signed PE in the other modality (see opposite sIntPE fMRI signals in Fig 9). Consequently, a conjunction analyses revealed no overlap.

Because the signed intensity PE effect is calculated by fitting a line through 3 predictive values (negative intensity PE meaning intensity lower than expected or positive intensity PE meaning intensity higher than expected, as compared to intensity as expected), the question arose whether the effect was constituted differently by negative or positive PEs. To ascertain this in an adjunct analysis, we set up a general linear model using both signed PEs as separate

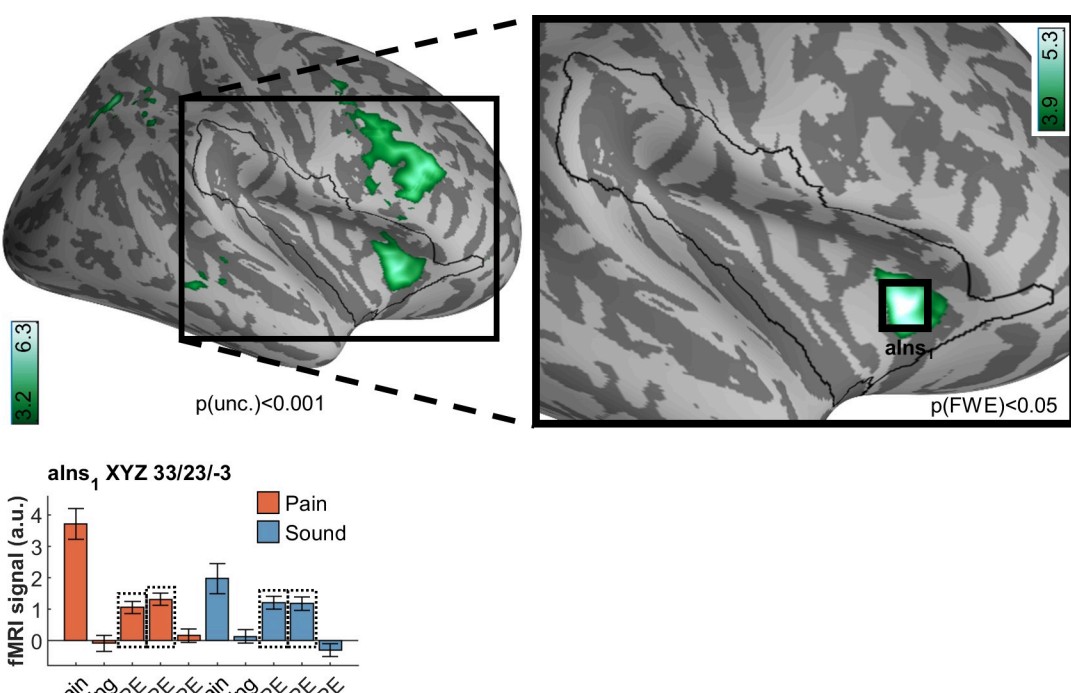

**Fig 8. Common brain activation associated with unsigned intensity and modality prediction errors.** The fMRI signal plot shows that the peak in the anterior insula (aIns$_1$) encodes PEs for every contrast included in the conjunction. Activations are overlaid on an average brain surface; for display purposes, activations in the whole brain lateral view are thresholded at p [uncorr.] < 0.001. The black line in the zoomed-in view delineates the region of interest and includes activations within the small volume FWE corrected at p[corr.] < 0.05. In the fMRI signal bar graph, black rectangles highlight the regressors used for analysis; solid line indicates analysis with the respective individual regressor, and dashed line indicates conjunction analysis. fMRI signal labels refer to the regressors used for each modality: "main" for main effects of modality, "ratings" for behavioral ratings, "modPE" for modality PEs, "uIntPE" for unsigned intensity PEs, and "sIntPE" for signed intensity PEs. Data used to produce the figure can be found at https://www.doi.org/10.17605/OSF.IO/7JBV3. fMRI, functional magnetic resonance imaging; FWE, family-wise error; PE, prediction error.

regressors; we then obtained the z-values of the pain and sound peak, respectively. For pain (XYZ$_{MNI}$ 36.4/−17.3/15.8), values were z = 2.582 (for negative intensity PE) and z = 3.053 (for positive intensity PE); for sound (XYZ$_{MNI}$ 49.4/−16.6/−13.4), values were z = 1.300 and z = 3.922, respectively. This indicates a comparable contribution of the negative and positive intensity PE component for pain, while sound activation more strongly driven by the positive intensity PE (louder than expected) component. Conjunction analyses combining negative and positive intensity PEs, performed separately for either modality, yielded p[uncorr.] = 0.00484 for pain, p[uncorr.] = 0.04468 for sound, which were not significant after correction for multiple comparisons.

In summary, the unsigned intensity PEs for pain and sound, as well as their modality PEs, strongly overlap in the anterior insula (Fig 6), whereas signed intensity PEs are accompanied by pain-dedicated activation in the dorsal posterior insula (Fig 9).

## Discussion

Using a Pavlovian learning paradigm with frequent reversals within and across aversive modalities in combination with SCR recordings and high-resolution fMRI, we were able to investigate signed and unsigned representations of PEs in the human brain. The data showed an unsigned representation of intensity PEs in the anterior insula indistinguishable for pain

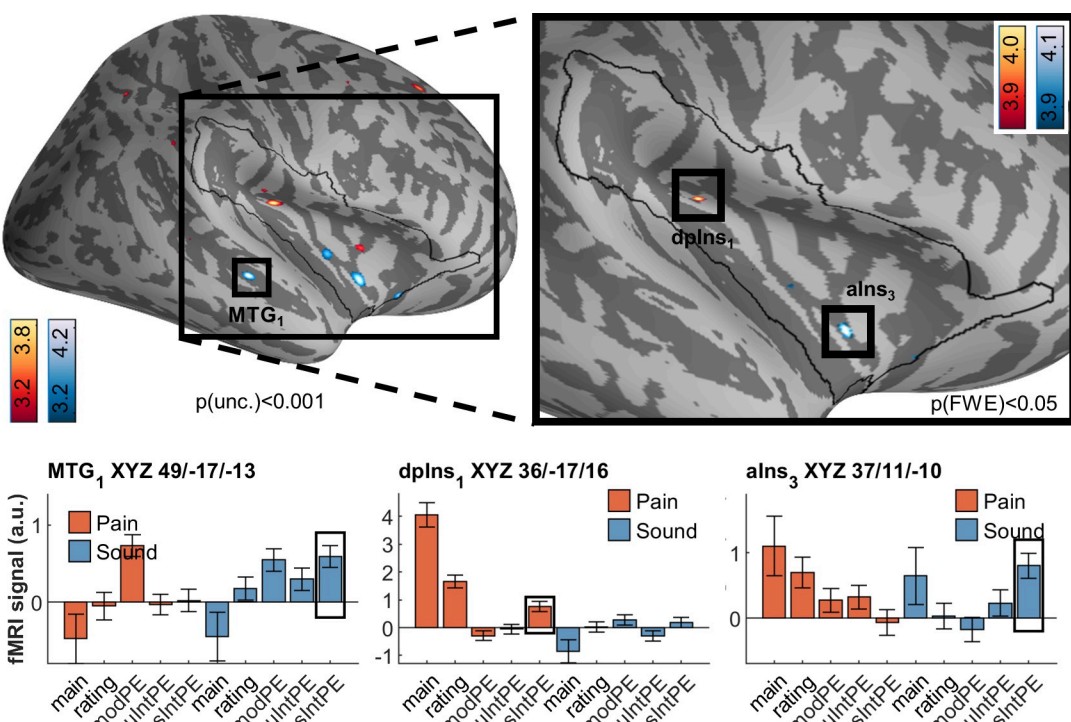

**Fig 9. Brain activation associated with by signed intensity prediction errors in pain (red/yellow) and sound (blue), including overlaps as per conjunction analyses (green).** Contrary to uIntPEs, circumscribed activation was detected for pain sIntPEs without any overlap with sound sIntPEs. Peak activation is located in the dorsal posterior insula (dpIns₁). For sound, several clusters in the anterior insula (e.g., aIns₃) were found, as well as middle temporal gyrus (MTG₁). Activations are overlaid on an average brain surface; for display purposes, activations in the whole brain lateral view are thresholded at p [uncorr.] < 0.001. The black line in the zoomed-in view delineates the region of interest and includes activations within the small volume FWE corrected at p[corr.] < 0.05. In the fMRI signal bar graphs, black rectangles highlight the individual regressors used for analysis. fMRI signal labels refer to the regressors used for each modality: "main" for main effects of modality, "ratings" for behavioral ratings, "modPE" for modality PEs, "uIntPE" for unsigned intensity PEs, and "sIntPE" for signed intensity PEs. Data used to produce the figure can be found at https://www.doi.org/10.17605/OSF.IO/7JBV3. fMRI, functional magnetic resonance imaging; FWE, family-wise error; PE, prediction error.

and aversive sounds, supporting a role of the anterior insula in coding unspecific arousal or salience. In addition, the same part of the anterior insula also strongly activated for PEs concerning stimulus modality. Most importantly, we could identify a circumscribed part of the dorsal posterior insula representing a signed PE for pain only, collocated with areas processing pain intensity per se.

A signed representation of an intensity PE for pain is a crucial teaching signal in reinforcement learning, as it is important to dissociate a low threat from a high threat stimulus. Such a representation for pain could plausibly be located in an area adjacent the anterior insula part representing unsigned intensity PEs and modality PEs. Alternatively, this representation could be located closer to representations of pain intensity: Coding of signed intensity PEs within areas coding for stimulus intensity per se was observed using a similar Pavlovian transreinforcer paradigm in the olfactory domain [26]. Indeed, our data show that a signed intensity PE for pain is represented in a part of the dorsal posterior insula [22,35]. Interestingly, we also identified a similar representation of a signed intensity PE for aversive sounds in or adjacent to primary auditory cortices [43,44], namely the middle temporal gyrus and temporal operculum. It also seems indicative of the more general involvement of the insula in pain perception [45] that the signed intensity PE in pain has little to none sound-related activation at all, whereas the signed intensity PE in sound includes some pain intensity-related activation.

We have replicated findings concerning pain-related activation in the dorsal posterior insula/parietal operculum and sound-related activation in the superior temporal gyrus [22]. Previously, these areas showed a clear main effect of pain and sound stimulation, respectively, but a crucial pain and sound rating-related increase in activation that is shallower or absent in nonnoxious intensities. In contrast to the previous study, we see a stronger correlation of the BOLD response to sound ratings, possibly owing to the higher intensities employed here.

Also, in agreement with previous studies, we observed an unsigned intensity PE for pain in the anterior insula [12,19,21]. The novel contribution is the fact that stimuli in different modalities (i.e., pain and aversive sounds) lead to the same activations in the anterior insula, with similar magnitudes. To our surprise, strong activation in the anterior insula was also observed for modality PEs (expect pain and receive sound and vice versa). fMRI signals for unsigned intensity PEs and modality PEs were very similar in magnitude. This disconfirms our hypothesis that at the level of the insula, modality PE carries less difference in salience between the expected and the real outcome, as compared to an unsigned intensity PE. Rather, it seems that surprise from unexpected sensory modalities is as much a source of anterior insula activation as from unexpected intensities. Notably, activation following modality PEs in either modality is characterized by the overlap with the other, with little differential involvement of structures dedicated to either modality or discriminating functions such as spatial orientation (also see S10 and S11 Figs). Instead, differences appear to be a matter of degree in the spread of activation, without substantial involvement of unimodally different structures. Our findings suggest that modality and unsigned intensity PEs are largely modality neutral and support findings that the anterior insula is richly interconnected part of the salience and attentional network involved in decision-marking, error recognition, and generally the guidance of flexible behavior [31,46–49]. Indeed, the large-scale activation following modality PEs and unsigned intensity PEs themselves does not correspond to any single network description, but seems to involve all of the above; possibly, different dynamics are at play over the course of the stimulation, which do not allow for the disentangling of single networks. In fact, recent meta-analytic evidence of resting-state functional connectivity points to the existence of a pain-related network centered on the anterior insula [50]. The activation associated with both pain-related (posterior insula) activation and that associated with PE-related (anterior insula) activation correspond well with connectivity gradients observed along the posterior–anterior axis [51–53].

Regions coding for aversiveness per se should exhibit overlaps in the respective rating-related activation across modalities. In the current data, this is the case, e.g., in the central operculum (with high proximity to the anterior insula) and—as per whole brain analysis—the anterior cingulate cortex (Fig 5B and 5C, S6 Fig). This overlap, while relatively sparse, is in line with previous results using similar supramodal paradigms [19,22,23] and correspond to known correlates of suffering [54]. As has been pointed out before [22,55], studies exhibiting a large modality-independent activation predominantly use comparatively brief stimuli [23,56], with functional imaging results potentially emphasizing the salience-/orienting-related activation. In the future, it could be worthwhile to consider remaining differences between the modalities, such as the focus on predicting target stimuli instead of passive perception. For example, spatial location is a parameter relevant for the painful stimuli only, and laterality effects or even stimulation more proximal to the ear could explain further nonspecific variance in the cerebral signatures.

Furthermore, as exemplified by the predominance of unsigned over signed PEs, aversiveness is confounded with salience: It is possible that the intentionally similar salience between the 2 inherently aversive modalities had overshadowed some modality-specific and supramodal mechanisms. It seems promising that future protocols include equisalient appetitive

conditions to tease out these mechanisms. For example, the involvement of different structures for reward and punishment has been demonstrated in studies using instrumental learning tasks [57,58] including intracranial recordings [30] or lesion studies [59]. In theory, the paradigm presented here also includes similar aspects, such as relief in the form of negative PEs (when more pain is expected, receiving less may be experienced as rewarding). Still, specifics of the protocol may have prevented more widespread overlaps between the modalities even though such supramodal parallels exist. Such specifics include a presumptive focus on contingencies as opposed to passive sensory perception or differences related to Pavlovian versus instrumental learning designs incorporating different decision-making processes, experiences of reward and punishment, and forms of feedback.

One of the strengths of the current paradigm include the parallel assessment of SCR, behavioral ratings for both expectation and outcome, as well as fMRI recordings which allowed us to investigate PEs in a multimodal fashion. Previous studies investigated PEs using cue-based pain paradigms [12,19,21,60]. In these paradigms, a cue predicts a stimulus intensity with a certain probability. However, the probability also determines the number of trials in which a PE occurs. This can lead to unbalanced designs in which certain PEs occur much more frequently than others. In addition, the fixed association of a specific cue with an outcome risks that specific features of the cue influence PE processing. Adopting a Pavlovian transreinforcer paradigm ameliorates these shortcomings and requires frequent relearning of contingencies and thus generates frequent PEs [25,26]. By defining a Markovian transition structure, we also controlled the nature of reversals; we confined our experiment to within-intensity/between-modality and between-intensity/within-modality reversals. Finally, introducing 2 CS in our task increased task difficulty. Even though we would have desired a more gradual learning curve for more fine-grained PEs, we have to attest a quick average learning performance (Fig 3B). Going forward, there are several options to increase task difficulty, such as using probabilistic instead of deterministic contingencies, adding intensities, or reducing discriminability.

We explicitly included expectation ratings, which allowed us to use the difference between the US and its expectation as a rating-derived PE [26]. Compared to model-derived PEs, this can account for within-subject differences in learning and can also capture PEs in erratic behaviors difficult to model in formal reinforcement learning models.

Although we aimed to perfectly match salience between stimulus modalities, high-intensity painful stimuli lead to higher SCR activation compared to low pain or either sound intensity (Fig 4), even though average SCR amplitudes between modalities were not statistically different. Technically, this is related to the fact that we were not able to increase sound pressure levels above a certain level [61] to avoid harm for the volunteers; this is a core obstacle when considering more sophisticated cross-modality matching procedures. However, the fMRI signal changes in the anterior insula for unsigned intensity PEs were similar for pain and sound, suggesting that the residual differences in SCR did not affect our results (Figs 6–8). In addition, previous accounts [62] have indicated that higher salience enhances memory performance. We tested this and observe no such effect: Learning performance did not substantially differ between any of the US groups (S14 Fig).

It is known that SCR predominantly shows arousal and similar effects, but is relatively insensitive concerning valence [27–29,63,64]. Here, SCR following unsigned or signed intensity PEs was little different from SCR following no PEs, while SCR following modality PEs was much higher. This might indicate that modality PEs provide a highly salient teaching signal even in the absence of intensity differences (S3 Fig).

Due to the task-inherent structure, signed pain intensity PEs can be correlated with actual pain ratings [57]. This collinearity can be remedied by orthogonalizing regressors in the general linear model used for fMRI analysis. However, this arbitrarily assigns the shared variance

to either of the 2 correlated regressors, depending on the order of the serial orthogonalization [65]. Therefore, we refrained from any orthogonalization in our analysis and thus only reveal areas that show unique variance tied to the regressors, including the signed intensity PEs for pain. This may have been a contributing factor to the relatively sparse activation following signed intensity PEs—this limitation could also be addressed by increasing task difficulty (see Discussion above).

At most, the clear spatial dissociation of intensity PEs for pain and sounds furthermore indicates a specificity of the signal; at least, it stands in marked contrast with the large overlap of activation for unsigned intensity and modality PEs in the anterior insula. Powerful learning models can utilize both a signed PE to update their predictions and an unsigned PE to update their learning rate [10,17,18]. Our results provide a neuronal basis for these models as we were able to reveal the simultaneous representation of both a signed and unsigned PE signal in spatially distinct regions of the insula.

In conclusion, our data provide clear evidence of anterior insula-centered, modality-independent unsigned PEs, not only concerning mismatched stimulus intensities across modalities, but also across sensory modalities themselves. Equally important, signed intensity PEs were associated with activation in or adjacent to sensory areas highly dedicated to unimodal processing. Neuronal data from both sources are the basis for reinforcement learning and further enhance our understanding of the functional synergies within the insula. Importantly, pathological learning mechanisms [1,9] and abnormalities in anterior insula-related function have been reported in chronic pain [50,66]. Our data therefore offers the possibility that a misrepresentation of PEs constitutes a potential mechanism in pain persistence.

## Materials and methods

The protocol conformed to the standards laid out by the World Medical Association in the Declaration of Helsinki and was approved by the local Ethics Committee (Ethikkommission der Ärztekammer Hamburg, vote PV4745). Participants gave written informed consent prior to participation and were aware of all aspects of the protocol except the randomized time point of reversal trials.

### Participants

A total of 47 healthy volunteers (sex: 26f:21m and age: 26.1 ± 4.5) were recruited through online advertisements (www.stellenwerk.de) and word of mouth. They were screened concerning study-specific and magnetic resonance imaging (MRI)-specific exclusion criteria as follows:

- age younger than 18, older than 40;

- insufficient visual acuity (correction with contact lenses only);

- conditions disqualifying for MRI scanners (e.g., claustrophobia or wearing a pacemaker);

- ongoing participation in pharmacological studies or regular medication intake (e.g., analgesics);

- analgesics use 24 hours prior to the experiment;

- pregnancy or breastfeeding;

- chronic pain condition;

- manifest depression (as per Beck Depression Inventory II, cutoff 14 [67]);

- somatic symptom disorder (as per Patient Health Questionnaire, cutoff 10 [68]);

- other neurological, psychiatric or dermatological conditions;

- inner ear conditions; and

- head circumference >60 cm (due to MRI scanner coil/headphone constraints).

Eligible participants were scheduled for a single lab visit. Experiments were conducted from October 2019 through March 2020. Statistics characterizing the sample are listed in S2 Table.

## Overview of the experiment

The sequence of measurements and timings of the protocol are displayed in Fig 1, while aspect pertaining to CS characteristics as well as contingencies are displayed in Fig 2. The experiment lasted about 2.5 hours. The experiment followed a full cross-over design, with every participant participating in all conditions. Participants learned associations of CS and US (painful heat or loud sound). These associations eventually changed in an unforeseeable manner and then had to be relearned. The experiment was run in a single visit, but split into 2 sessions to reduce participant fatigue and carry-over effects. Prior to the experimental sessions, participants were calibrated according to their pain and sound sensitivity. At the start and the end of the experiment, participants filled out psychological questionnaires outside the scanner. Electrodermal activity was measured throughout the experimental sessions.

## Unconditioned stimuli

Heat stimuli were delivered using a CHEPS thermode (Medoc, Ramat-Yishai, Israel) attached to the volar forearm. Basic stimulus parameters included a 32°C baseline temperature and 10°C/s rise and fall rates. Sound stimuli were delivered using MRI-compatible headphones (MR confon, Magdeburg, Germany). A pure sound (frequency 1,000 Hz, sampling rate 22,050 Hz) was generated during runtime using MATLAB.

## Calibration of unconditioned stimulus intensities

Prior to the experiment proper, participants underwent US calibration to determine 2 intensities at VAS 25 and VAS 75 for both modalities (heat and sound). During the experiment, only these 4 stimuli were used. All stimuli lasted 3 seconds at plateau, except for four 10-second long, low-intensity preexposure stimuli used for familiarization and preheating of the skin.

Heat and sound stimuli were presented and rated in an analogous fashion. Like in a previous study comparing neuronal responses to the 2 modalities [22], we used the descriptor "painfulness" for heat, while we used the descriptor "unpleasantness" for sound. After calibration, all stimuli were above the respective pain and unpleasantness thresholds and were therefore displayed on simple 0 to 100 VASs for both modalities.

For heat, anchors were displayed for "minimal pain" (0) and "unbearable pain" (100). Pain was defined as the presence of sensations other than pure heat intensity, such as stinging or burning [69].

For sound, participants were instructed to rate between anchors labeled "minimally unpleasant" (0) and "unbearably unpleasant" (100). Unpleasantness was defined as a bothersome quality of the sound emerging at a certain loudness.

During the calibration procedure performed in the running MRI scanner, 2 stimulus intensities each were obtained for the heat and sound modality (low/high pain and low/high noise). Heat stimuli ranged from 43 to 49°C, and sound stimuli ranged from 89.1 through 103.0 dBA. Calibration was constrained such that participants had to reach a certain

- minimum physical intensity (43°C for heat, 20% system volume for sound, $n = 1$ received 10%) and

- minimum physical difference between the VAS 25 and 75 stimuli (1.5°C for heat, 15% system volume for sound; $n = 1$ received 1°C, $n = 8$ received 10%).

If either condition was not met, physical intensities were automatically adjusted to the minimum (e.g., if participant reported VAS 25 for 41°C, temperature was raised to 43°C). Furthermore, to ensure discriminability within stimulus modalities, participants had the calibrated US played back to them and were explicitly asked 3 questions, namely that both intensities of the respective modality

- were painful (for heat) or unpleasant (for sound);

- were perspectively tolerable throughout repeated trials in 2 sessions; and

- were easily discriminable.

If either question was answered in the negative, the calibrated intensities were adjusted, but never below the minimum requirements listed above.

## Learning protocol

Learning the CS–US associations was designed as a Pavlovian transreinforcer reversal learning task [25,26]. Two CS would independently predict one of 4 US, namely 2 intensities of painful heat and 2 intensities of unpleasant sound. Participants were presented with one of the 2 CS (Fig 2C and 2D) and then asked to choose which of the 4 US they believed to be preceded by it (symbols in Fig 2B). After making their choice, they would actually be exposed to one of the 4 US (see Fig 1C for trial structure). If they were correct, no further learning was required; if not, they would have the opportunity to learn the correct association for the next occurrence of the CS. They would then rate their pain or unpleasantness on a 0 to 100 VAS, as during US calibration. Both CS signified an independent sequence of associations with the US. Both CS were randomly drawn for each participant from a library of 8 fractal pictures generated using the SHINE toolbox in MATLAB (Fig 2A). Which of the 2 CS was presented in each trial was fully randomized, as were the US for the respective initial associations, and the display order of the US prediction rating.

Crucially, after a number of trials with deterministic CS–US association, the association underwent an unannounced reversal either in terms of intensity (previously low US intensity would now be high or vice versa), or modality (previous pain US would now be a sound US or vice versa) (Fig 2C and 2D). The number of trials that an association was upheld was randomly determined from [3, 3, 4, 5] (i.e., 3.75 trials on average). After each reversal, participants therefore made an error in predicting the following US and subsequently had to learn the new association. As reversals on both dimensions were precluded, each session included 8 reversals per CS to cover all possible reversals. Task performance was assessed by the percentage of correct predictions.

## Psychological questionnaires

Prior to and immediately after the experiment, participants filled out several questionnaires assessing state and trait psychological constructs. These are listed in S2 Table alongside statistics characterizing the sample.

## Psychophysiological recordings

Electrodermal activity was measured with MRI-compatible electrodes on the side of the left hand opposite the thumb. Electrodes were connected to Lead108 carbon leads (BIOPAC Systems, Goleta, California, United States of America). The signal was amplified with an MP150 analog amplifier (also BIOPAC Systems). It was sampled at 1,000 Hz using a CED 1401

analog-digital converter (Cambridge Electronic Design, Cambridge, United Kingdom) and downsampled to 100 Hz for analysis.

Analysis was performed using the Ledalab toolbox for MATLAB [70]. Single participant data were screened for artifacts that were removed if possible by using built-in artifact correction algorithms. Of 47 participants, 1 was excluded due to equipment malfunction and 9 due to skin conductance nonresponsiveness. From the remaining 37 participants, a total of 101 of 6,016 segments (1.7%) were excluded due to unsalvageable artifacts. Using a deconvolution procedure, we computed the driver of phasic skin conductance (SCR). Stimulus phase response windows were offset between the 2 stimulus modalities [22]—we attribute an earlier onset following acoustic stimulation to reduced latency from the delivery system and neuronal transmission. To determine response windows, we obtained the times for average peaks of the respective modality and selected the data range ± 1.25 seconds: For pain, response windows were set between 2.42 seconds and 4.92 seconds and between 1.15 seconds and 3.65 seconds for sound. SCR segments were log- and z-transformed within participants to reduce the impact of intra- and interindividual outliers [27]. Subsequently, segments were averaged within participants for several conditions corresponding to the behavioral performance of participants (e.g., intensity PE following low painful stimulation, or high painful stimulation). SCR was used because it is an objective measure of general sympathetic activity and therefore a measure of arousal, stimulus salience, and several associated psychological processes [27,28,63,71,72]. It is routinely used in assessing painful [12,22,73] as well as acoustic stimulation [74].

## fMRI acquisition and preprocessing

Functional and anatomical imaging was performed using a PRISMA 3T MRI Scanner (Siemens, Erlangen, Germany) with a 20-channel head coil. An fMRI sequence of 56 transversal slices of 1.5-mm thickness was acquired using T2*-weighted gradient echo-planar imaging (EPI; 2001 ms TR, 30 ms TE, 75° flip angle, $1.5 \times 1.5 \times 1.5$ mm voxel size, 1-mm gap, $225 \times 225 \times 84$ mm field of view, simultaneous multislice imaging with a multiband factor of 2, and an acceleration factor of 2 with generalized autocalibrating partially parallel acquisitions reconstruction). Additionally, a T1-weighted MPRAGE anatomical image was obtained for the entire head (voxel size $1 \times 1 \times 1$ mm, 240 slices).

For each participant, fMRI volumes were realigned to the mean image in a 2-pass procedure and nonlinearly coregistered to the anatomical image using the CAT12 toolbox for SPM (Christian Gaser and Robert Dahnke, http://www.neuro.uni-jena.de/cat). In short, this novel nonlinear coregistration segments both the mean EPI and the T1 weighted image and performs a nonlinear spatial normalization of the segmented tissue classes from the mean EPI using the segmented tissue classes from the T1 scan as a template. Finally, individual brain surfaces were generated, using CAT12.

## General statistical approach

Unless otherwise noted, analyses except the fMRI analyses were performed using linear mixed models with random intercept using trial-by-trial parameters. In the case of mixed (within/between) descriptive statistics, standard errors were calculated using the Cousineau–Morey approach [75]. The significance level for analyses of behavioral and psychophysiological data was set to $p = 0.05$.

## Analysis of imaging data

Subject-level analyses were performed on the 3D (volume) data in native space without smoothing, as required for surface mapping. We computed a general linear model with a

canonical response function to identify brain structures involved in the processing of each stimulus modality and corresponding to various predictions and PEs inherent in the protocol. Realignment (motion) parameters were included as nuisance variables, to further mitigate motion-related artifacts.

A general linear model was set up with one regressor for stimulus main effects in each modality (heat or sound), using onsets of the US and a 3-second boxcar convolved with the canonical HRF. Furthermore, we have added a parametric modulator each for pain or unpleasantness (using behavioral ratings). An additional 3 parametric modulators for each modality were entered for modality PEs and intensity PEs. Hence, the analysis included 10 regressors in total, which are labeled in the fMRI signal bar graphs Figs 5–9, for each modality, as "main" for main effects of modality, "rating" for behavioral ratings, "modPE" for modality PEs, "uIntPE" for unsigned intensity PEs, and "sIntPE" for signed intensity PEs. Modality PEs were entered unsigned due to their nonparametric nature, whereas intensity PEs were entered both unsigned (absolute) and signed. All parametric modulators were z-scored within participants and sessions. In either model, global or sequential orthogonalization between regressors were turned off to preserve only the unique (nonshared) variance components [26,65]. This approach allows for the interpretation of consecutively entered parametric modulators even if correlations to previous regressors exist.

We opted for surface-based analyses of fMRI data to enhance discrimination between modalities processed in adjacent brain regions [22]; for an example of pseudo-overlap detected across the Sylvian fissure, see S5 Fig (row 3), particularly in slices −28 through −16. Results from subject-level analyses were mapped to brain surfaces obtained via the CAT12 segmentation procedure. The mapped subject-level results were then resampled to correspond to cortical surface templates, and smoothed with a 6-mm full width half maximum 2D kernel. Group-level within-subjects analyses of variance were performed including the mapped contrasts. The original, unmapped contrasts were used for volume-based group-level analyses to assess subcortical activation. Volume results were then warped using DARTEL normalization and smoothed with a 6-mm full width-half maximum 3D kernel. Volume-based results are provided in the Supporting information and referenced where relevant.

Contrasts employed for any of the analyses were either performed against low-level baseline (e.g., Pain > 0), as a conjunction of a differential modality contrast and one against low-level baseline (e.g., Pain > Sound ∧ Pain > 0), or as a conjunction of both modalities (e.g., Pain ∧ Sound).

### Regions of interest and statistical correction of imaging results

As laid out above and because pain is the modality of interest in this study, we focused the analyses on the contralateral (right) periinsular cortices as regions of interest used for small volume correction of significance level [12,19,22]. The region of interest included the entire insular cortex (dorsal hypergranular, dorsal granular, dorsal dysgranular, dorsal agranular ventral dysgranular/granular, and ventral agranular), as well as dorsally adjacent areas of the parietal operculum (A40rv), central operculum (A1/2/3ll, A4tl), and frontal operculum (A44op, A12/47l). It was created using the Human Brainnetome Atlas [76]. Results were considered after correction for family-wise error rate of $p < 0.05$ within the region of interest (denoted p [corr.]) or after correction for whole brain/all vertices (denoted p[corr. wb.]), unless otherwise noted. No extent threshold was used.

### Supporting information

**S1 Table. Effects of modality and intensity on SCRs, by PE type.** Parameters obtained from linear mixed models with random subject intercept. Differences between the conditions are

largest in trials with no PE and smallest in trials with modality prediction error (cf. S1 Fig). Data used to produce the table can be found at https://www.doi.org/10.17605/OSF.IO/7JBV3. PE, prediction error; SCR, skin conductance response.
(DOCX)

**S2 Table. Sample characteristics.** Data used to produce the table can be found at https://www.doi.org/10.17605/OSF.IO/7JBV3. BDI-II, Beck Depression Inventory II; exp., experiment; FPQ, Fear of Pain Questionnaire; MDMQ, Multidimensional Mood Questionnaire; PHQ15, Patient Health Questionnaire-15; PRSS, Pain-Related Self-Statements; PSQ, Pain Sensitivity Questionnaire; PVAQ, Pain Vigilance and Awareness Questionnaire; STAI, State-Trait Anxiety Inventory.
(DOCX)

**S1 Fig. Illustration of reversal types.** Both CS have an independent sequence of deterministic associations with one of the 4 US (also see Fig 2). The dashed lines illustrate reversals for CS1 (black) or CS2 (white). First column, CS2 intensity reversal from low to high heat; second column, CS1 intensity reversal from low to high sound; third column, CS2 modality reversal from low heat to low sound; fourth column, modality reversal from high sound to high heat. Data used to produce the figure can be found at https://www.doi.org/10.17605/OSF.IO/7JBV3. CS, conditioned stimuli; US, unconditioned stimuli.
(TIF)

**S2 Fig. Behavioral results for low and high unconditioned pain and sound stimuli. (A)** Calibrated stimulus intensities corresponding to VAS 25 (low intensity) and VAS 75 (high intensity) for pain stimuli and sound stimuli. Each line represents the 2 intensities per modality per participant; the violin plots aggregate over participants. **(B)** Single trial ratings following pain stimulation and sound stimulation. Every column represents a single participant's response to the respective intensity and modality; the bordered circle is a participant's mean rating. The gray dashed lines is the "intended" rating as per calibration (VAS 25 for low and VAS 75 for high intensities). The black line is the actual mean rating over all participants. Data used to produce the figure can be found at https://www.doi.org/10.17605/OSF.IO/7JBV3. VAS, visual analogue scale.
(TIF)

**S3 Fig. Results from SCR measurements, by PE type.** Rows show group means of SCR following no PE (row 1), intensity PE (row 2), and modality PE (row 3). Column show poststimulus SCR (left) and SCR averaged within the indicated response windows (right).Differences between conditions are largest in the no PE condition, smallest in the modality PE condition, which also shows the largest SCR amplitudes. Statistics of differences between conditions are displayed in S1 Table. All plots are based on log- and z-transformed data. Data used to produce the figure can be found at https://www.doi.org/10.17605/OSF.IO/7JBV3. PE, prediction error; SCR, skin conductance response.
(TIF)

**S4 Fig. Lateral and medial views of brain surface results for heat onsets (yellow/red), sound onsets (blue), and their conjunction (green).** Activations are overlaid on an average brain surface and thresholded at p[uncorr.] < 0.001. The black line delineates the region of interest whose results are highlighted in Fig 5A and 5B. Data used to produce the figure can be found at https://www.doi.org/10.17605/OSF.IO/7JBV3. L, left hemisphere; R, right hemisphere.
(TIF)

**S5 Fig. Brain volume results for heat onsets (yellow/red), sound onsets (blue), and their conjunction (green).** Activations are overlaid on an average brain volume and thresholded at p[uncorr.] < 0.001.
(TIF)

**S6 Fig. Lateral and medial views of brain surface results for pain ratings (yellow/red), sound ratings (blue), and their conjunction (green).** Activations are overlaid on an average brain surface and thresholded at p[uncorr.] < 0.001. The black line delineates the region of interest whose results are highlighted in Fig 5B and 5C. Data used to produce the figure can be found at https://www.doi.org/10.17605/OSF.IO/7JBV3. L, left hemisphere; R, right hemisphere.
(TIF)

**S7 Fig. Brain volume results for pain ratings (yellow/red), sound ratings (blue), and their conjunction (green).** Activations are overlaid on an average brain volume and thresholded at p[uncorr.] < 0.001.
(TIF)

**S8 Fig. Lateral and medial views of brain surface results for unsigned intensity prediction errors for heat (yellow/red), sound (blue), and their conjunction (green).** Activations are overlaid on an average brain surface and thresholded at p[uncorr.] < 0.001. The black line delineates the region of interest whose results are highlighted in Fig 6. Data used to produce the figure can be found at https://www.doi.org/10.17605/OSF.IO/7JBV3. L, left hemisphere; R, right hemisphere.
(TIF)

**S9 Fig. Brain volume results for unsigned intensity prediction errors for heat (yellow/red), sound (blue), and their conjunction (green).** Activations are overlaid on an average brain volume and thresholded at p[uncorr.] < 0.001.
(TIF)

**S10 Fig. Lateral and medial views of brain surface results for modality prediction errors for heat (yellow/red), sound (blue), and their conjunction (green).** Activations are overlaid on an average brain surface and thresholded at p[uncorr.] < 0.001. The black line delineates the region of interest whose results are highlighted in Fig 7. Data used to produce the figure can be found at https://www.doi.org/10.17605/OSF.IO/7JBV3. L, left hemisphere; R, right hemisphere.
(TIF)

**S11 Fig. Brain volume results for modality prediction errors for heat (yellow/red), sound (blue), and their conjunction (green).** Activations are overlaid on an average brain volume and thresholded at p[uncorr.] < 0.001.
(TIF)

**S12 Fig. Lateral and medial views of brain surface results for signed intensity prediction errors for heat (yellow/red) and sound (blue).** No significant conjunction activation prevails. Activations are overlaid on an average brain surface and thresholded at p[uncorr.] < 0.001. The black line delineates the region of interest whose results are highlighted in Fig 9. Data used to produce the figure can be found at https://www.doi.org/10.17605/OSF.IO/7JBV3. L, left hemisphere; R, right hemisphere.
(TIF)

**S13 Fig. Brain volume results of signed intensity prediction errors for heat (yellow/red) and sound (blue).** No significant conjunction activation prevails. Activations are overlaid on an average brain volume and thresholded at p[uncorr.] < 0.001.
(TIF)

**S14 Fig. Mean performance split by modality/intensity.** Grand mean performance is shown in Fig 3B. Data used to produce the figure can be found at https://www.doi.org/10.17605/OSF.IO/7JBV3.
(TIF)

## Acknowledgments

We thank Thorsten Kahnt for comments and scripts concerning the randomization procedure; Saša Redžepović for providing scripts used for CS fractal generation; Jürgen Finsterbusch, Katrin Bergholz, Waldemar Schwarz, and Kathrin Wendt for technical assistance during MRI data collection; and Alina Schaefer and Jannis Petalas for their assistance with data collection.

## Author Contributions

**Conceptualization:** Björn Horing, Christian Büchel.

**Data curation:** Björn Horing.

**Formal analysis:** Björn Horing, Christian Büchel.

**Funding acquisition:** Christian Büchel.

**Investigation:** Björn Horing.

**Methodology:** Björn Horing, Christian Büchel.

**Project administration:** Christian Büchel.

**Resources:** Christian Büchel.

**Software:** Björn Horing.

**Supervision:** Christian Büchel.

**Visualization:** Björn Horing.

**Writing – original draft:** Björn Horing.

**Writing – review & editing:** Björn Horing, Christian Büchel.

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
