## [Editor Report · Decision Letter 0]

12 Jan 2022

Dear Dr Horing, 

Thank you for submitting your manuscript entitled "Pain-related learning signals in the human insula" for consideration as a Research Article by PLOS Biology.

Your manuscript has now been evaluated by the PLOS Biology editorial staff, as well as by an academic editor with relevant expertise, and I am writing to let you know that we would like to send your submission out for external peer review. Please accept my apologies for the delay in communicating this decision to you.

Once your full submission is complete, your paper will undergo a series of checks in preparation for peer review. Once your manuscript has passed the checks it will be sent out for review. To provide the metadata for your submission, please Login to Editorial Manager (https://www.editorialmanager.com/pbiology) within two working days, i.e. by Jan 14 2022 11:59PM.

If your manuscript has been previously reviewed at another journal, PLOS Biology is willing to work with those reviews in order to avoid re-starting the process. Submission of the previous reviews is entirely optional and our ability to use them effectively will depend on the willingness of the previous journal to confirm the content of the reports and share the reviewer identities. Please note that we reserve the right to invite additional reviewers if we consider that additional/independent reviewers are needed, although we aim to avoid this as far as possible. In our experience, working with previous reviews does save time. 

If you would like to send previous reviewer reports to us, please email me at ggasque@plos.org to let me know, including the name of the previous journal and the manuscript ID the study was given, as well as attaching a point-by-point response to reviewers that details how you have or plan to address the reviewers' concerns. 

Given the disruptions resulting from the ongoing COVID-19 pandemic, please expect some delays in the editorial process. We apologise in advance for any inconvenience caused and will do our best to minimize impact as far as possible.

Kind regards,

Gabriel

Gabriel Gasque

Senior Editor

PLOS Biology

ggasque@plos.org

---

## [Decision Letter · Decision Letter 1]

22 Feb 2022

Dear Dr Horing,

Thank you for submitting your manuscript "Pain-related learning signals in the human insula" for consideration as a Research Article at PLOS Biology. I'm handling your paper temporarily while my colleague Dr Gabriel Gasque is out of the office. Your manuscript has been evaluated by the PLOS Biology editors, an Academic Editor with relevant expertise, and by two independent reviewers. A third reviewer had been recruited, but failed to return their comments in a timely fashion.

In light of the reviews (below), we are pleased to offer you the opportunity to address the comments from the reviewers in a revised version that we anticipate should not take you very long. We will then assess your revised manuscript and your response to the reviewers' comments and we may consult the reviewers again.

We expect to receive your revised manuscript within 1 month.

**IMPORTANT - SUBMITTING YOUR REVISION**

*Resubmission Checklist*

*Published Peer Review*

*PLOS Data Policy*

*Blot and Gel Data Policy*

Sincerely,

Roli Roberts

Roland G Roberts PhD

Senior Editor

PLOS Biology

on behalf of

Gabriel Gasque

Senior Editor

PLOS Biology

ggasque@plos.org

REVIEWERS' COMMENTS:

Reviewer #1:

In this manuscript, the authors present the results of a brain imaging study examining the brain representation of two types of prediction errors : 1) prediction errors pertaining to the intensity of the stimulation, and 2) prediction errors related to the modality of the stimulation (auditory or thermal). In order to generate these two types of prediction errors, they designed a very clever and elegant experiment where the outcomes predicted by predictive cues could either change on the intensity dimension (from low to high or from high to low), or on the modality dimension (from auditory to thermal or thermal to auditory). I didn't notice any important flaws in study design or analyses. Since this is to my knowledge the first study investigating modality prediction error, I would recommend publication. Still I have a relatively long list of minor issues, which I think should be addressed.

1. To me, the key aspect of the study are the modality prediction errors. This should be made clearer in the introduction. More specifically, why is it important to study these modality prediction errors? Are there concrete examples where the modality of the stimulation abruptly switches like in their experiment? 

2. Related to the above point, I don't understand why the authors make the prediction of a weaker PE for switches in modality « in this case we expected a weaker signal, as the intensity - and thus salience and other general aspects - are intendedly not different between the expected and the received US. ». Modality PEs represent a qualitative change… I thought they would be even more surprising… That being said, the probability of intensity or modality changes is the same. Based on that, I wouldn't expect much variability in the strength of the PE. But I don't think that the amplitude of the PE is the right way to frame the hypotheses. What is more interesting are the structures involved. In the case of modality, there is the possibility that the PE could involve sensory and spatial orientation systems.

3. The rationale for focusing on the insula is also not clear. It feels like it would bias the results to only reveal unsigned and modality-independent PEs. It also produces an asymmetry between auditory (processed in STG, which is not in the mask) and thermal stimuli (processed in posterior insula and parietal operculum, which are within the mask). If I personally had to choose a priori regions, I would have focused on 1) "salience" regions (aINS AND ACC), 2) sensory regions (OP, dpINS, S1, STG), and 3) spatial orientation fronto-parietal regions. I have no doubts that a case can be made for the entire insula, but the paper would be stronger if the authors could present a brief theory of insula function in the introduction.

4. I also think that I would have preferred an introduction that leaves more room to the possibility of signed, not just unsigned, prediction errors. On the other hand, I also understand the need to tailor the introduction to the main findings of the experiment. Still, I think I would have preferred an introduction that raises the possibility of signed prediction errors, and then a discussion as to why they perhaps didn't came out as strongly as expected? This is more a remark than a criticism. I leave that entirely up to the authors.

5. Modality PEs are confounded with spatial location. I don't think that this is a problem here, but I think it should be discussed briefly. It could be eventually nice to have an experiment that tests spatial location within-modality.

6. I don't think that subjective intensities were perfectly matched. "100 = unbearable pain" probably refers to an absolutely stronger sensation than "100 = extremely unpleasant". Cross-modality matching would have been a better psychophysical technique here. But as I said previously, I don't think that the magnitude of the PE is particularly important here. This is more a remark than a criticism. The authors should simply be careful not to over-state the matching in absolute intensity.

7. There is a potential problem with the use of parametric modulators to track PEs. For instance, the parametric modulator for unsigned intensity PE (-1: lower than expected; 0: no change; 1: higher than expected) could be - for instance - entirely driven by a very strong difference between higher than expected and no change, with no difference between smaller than expected and no change. A stricter approach would have be to look at the conjunction of (lower than expected > no change) and (higher than expected > no change). The same principle applies to modality PEs and signed PEs, and skin conductance responses. I think the findings would be stronger if the authors could show that the same findings hold when using this stricter criterion. However, I am willing to accept the parametric modulators if this is only what comes out as significant.

8. Prediction errors were mostly (only ?) observed for the first « reversal » trial. This is not a problem in and of itself, but I think that this might have reduced the power to observe statistically significant effects. Perhaps a better approach would have been to have the transitions more probabilistic, rather than all-or-none. This is just a remark, not necessarily a problem that the authors need to assess in their manuscript. 

9. It seems like SCR from all participants were lumped together as if originating from one very long experiment in a single subject (or in a « fixed effects » manner). Ideally, the authors should have used summary statistics from every participant, or even better a hierarchical regression model taking into consideration within and between-subjects effects. The problem is that « fixed effects » can be driven by a few participants and not reflect group effects. The paper would be stronger if the results came out as significant using an approach that takes into consideration between-subjects variability. That being said, analyses of SCRs could be an exception to this general principle given the difficulties in dealing with non-responders. If that is the case, maybe the authors could justify this methodological choice a little bit better in the methods section. What stroke me as a bit odd were the relatively high p-values, which I think should be interpreted with caution. Note that I don't think that SCR effects are a « sine qua non » condition for interpreting brain results. Arguably, SCRs could be more noisy than brain signals. But I agree that starting with SCR results tell a better story.

10. Figure 4D indicates that SCRs for modality and intensity PEs have the same magnitude. This is contradictory with what can be seen in figure 4C. Why the discrepancy? 

11. First paragraph of imaging results: I would have expected the authors to report STG findings in the text as well.

12. Why call the effects of the stimulus « onset » in the figures. Was only the onset of the stimulations modelled?

13. I found the discussion to be too defensive and focusing on methodological considerations. It would be preferable to at least start by explaining why the key findings of the experiment are interesting. This parallels my comments on the introduction.

Despite my perhaps long list of comments, I'd like to re-state that this is a well-designed and well-conducted study. Findings of unsigned modality-independent PEs in the aINS are interesting and novel. Examining modality-related PEs is novel and original, and the paper is likely to get cited because of that. There still are a few imperfections (see above) - notably experimental effects might not have always come out as strongly as desired - but that doesn't undermine the originality of the study and its main findings. The introduction and discussion could be substantially improved, but this is well within the author's reach.

Reviewer #2:

Major comments.

1. To distinguish aversiveness from saliency, I wondered whether the experimental design could have included an appetitive condition. Could this be discussed in comparison to studies with different designs comparing reward from punishment learning signals since they also report a critical role of the anterior insula during punishment avoidance learning and prediction errors signals in fMRI (Pessiglione et al., Nature 2006), intracranial recordings (Gueguen et al., Nat. Comm 2021) and lesion studies (Palminteri et al., Neuron 2011). 

2. Relatedly, although there is a paragraph commenting study design difference between probabilistic RL tasks and the pavlovian approach used in this study, I think further discussing the advantages of the proposed paradigm (with binary signals) over parametric designs (with more graded expectation/outcome values) to study the PE. The entire paper is framed on the concept of PE but the experiment seems not optimal to examine fine-grained PE signals (signed and unsigned PE signals which were sub-sampled by the experimental structure) or even to check that the PE signals are "truly prediction error signals" according to a formal/axiomatic approach. I might have missed critical aspects of the paradigm or in the statistical analyses made (see below).

3. I could not figure out exactly what was done in terms of fMRI data analyses: in the figures 5-9, authors label their regressors (VAS, PE, modPE, modality PE (repeated twice in fig5 legend, please correct, …). I could not find a clear description of these regressors so that I had to guess what was done exactly (there are 7 lines of text page 23 to explain what was done to generate figs 5, 6, 7, 8, 9). Relatedly, I could not find which time window within the task was used to perform the fMRI analyses (the US period?). For instance, when investigating VAS (stimulus perception), fMRI could be either triggered by the US stimulus or by the VAS rating onset.

4. Unpleasantness/Pain ratings: in the paper it seems that unpleasantness signals are domain specific (activating modality specific brain regions); other groups report modality independent unpleasantness signals, notably in the anterior insula (e.g., Corradi del Acqua et al 2016) ?. This issue (modality specific vs. independent of unpleasantness) could be discusses. I also wondered whether using "only" four types of unpleasant stimuli (low vs. high pain and low vs. high sound) could be an issue compared to paradigms generating perhaps more variable ratings? 

5. Minor points: 

5.1 I would also be interested if the authors could articulate clearly the results of this study and their own previous work in which they conclude that anterior insula pain expectation /PE signals are not related to aversiveness (Fazeli & Buchel 2018). 

5.2 In the behavioral analyses, I suggest to use a more classic ANOVA followed by post-hoc test instead of using multiple t-tests to compare the conditions 

5.3 could you check throughout the manuscripts the terminology chosen to explain each concept and check for consistency across sections to reduce possible reader's confusion (e.g., for VAS pain/unpleasantness ratings, these are interpreted as a proxy of "stumulus perception" and/or "intensity effects"; 

5.4 Figures legends: please define the meaning of black (continuous vs. dashed lines) rectangles (does this corresponds to significant regerssors ?)

---

## [Editor Report · Decision Letter 2]

1 Apr 2022

Dear Drs Horing and Büchel, 

Thank you for submitting a revision of your manuscript entitled "Pain-related learning signals in the human insula" for consideration as a Research Article by PLOS Biology. As with all papers reviewed by the journal, yours was evaluated by the PLOS Biology editors as well as by an Academic Editor with relevant expertise. In this case, the Academic Editor felt comfortable looking over the revisions that were made, so the independent reviewers were not consulted in this second round. 

Based on our editorial evaluation and the advice of the Academic Editor, we will probably accept this manuscript for publication, provided you satisfactorily address the the writing, data and other policy-related requests below.

In terms of writing changes, we ask that you consider a modified title and we suggest one of the following:

“Modality-independent and pain-specific learning signals in the human insula”

OR

“The human insula processes both modality independent and pain-specific learning signals”

We also ask that you spend some time working on the abstract of your paper to ensure it is more broadly accessible to the wider PLOS Biology audience. We've provided a suggested a modification of your abstract below to give you a sense of the type of thing we'd be looking for.

-----

Example:

Prediction errors (PEs) are generated when there are differences between the expected and the actual outcome of an event. The insula is a key brain region involved in pain processing, and studies have shown that the insula responds to PEs that indicate an unexpected outcome has occurred (unsigned PEs). In addition to signaling general magnitude information, PEs can give specific information on the direction of this change – whether better or worse than expected. Whether the unsigned PE responses in the insula are pain-specific or reflective of a more processing of aversive events irrespective of modality, and whether the insula can process signed PEs at all, is unclear. Understanding these specific mechanisms has implications for understanding how pain is processed in the brain in both health and in chronic pain conditions. Here, using a study design in which 47 subjects learned associations between two conditioned stimuli (CS) with four unconditioned stimuli (US; painful heat or loud sound, of one low and one high intensity each) while undergoing functional magnetic resonance imaging (fMRI) and skin conductance response (SCR) measurements, we demonstrate that activation in the anterior insula correlated with unsigned intensity PEs, irrespective of modality. Conversely, signed intensity PE signals were modality-specific, with signals from pain but not sound signed PEs located in the dorsal posterior insula. Previous studies have identified abnormal insula function and abnormal learning as potential causes of pain chronification. Our findings link these results and suggest a misrepresentation of learning relevant prediction errors in the insular cortex may serve as an underlying factor in chronic pain.

-----

Below my signature you will see information on all of the additional data and policy information that we require before we can move on to an acceptance of this manuscript. As you address these items, please also take this last chance to review your reference list to ensure that it is complete and correct. If you have cited papers that have been retracted, please include the rationale for doing so in the manuscript text, or remove these references and replace them with relevant current references. Any changes to the reference list should be mentioned in the cover letter that accompanies your revised manuscript.

We expect to receive your revised manuscript within two weeks. 

*Published Peer Review History*

*Press*

Please do not hesitate to contact me should you have any questions. I look forward to hearing back from you and moving forward with this manuscript at PLOS Biology.

Sincerely,

Kris

Kris Dickson, Ph.D.

Neurosciences Senior Editor/Section Manager

kdickson@plos.org

PLOS Biology

Fig. 2 C, D; Fig. 3 A, B; Fig. 4 A-D; Fig. 5B; graphs embedded in Fig 6-9; Fig. S1; Fig. S2 A, B; Fig S3 and Fig. S14.

Please also ensure that figure legends IN YOUR MANUSCRIPT include information on WHERE THE UNDERLYING DATA CAN BE FOUND, and ensure your supplemental data file/s has a legend.

***Please also make the data that you are going to deposit in public databases publicly available now. THIS INFORMATION NEEDS TO BE PROVIDED NOW AS IT IS REQUIRED BEFORE FINAL ACCEPTANCE CAN OCCUR. Please also provide the accession codes.

---

## [Editor Report · Decision Letter 3]

15 Apr 2022

Dear Björn and Christian,

On behalf of my colleagues and the Academic Editor, Chris Summerfield, I am pleased to say that we can in principle accept your Research Article "The human insula processes both modality-independent and pain-selective learning signals" for publication in PLOS Biology, provided you address any remaining formatting and reporting issues. These will be detailed in an email that will follow this letter and that you will usually receive within 2-3 business days, during which time no action is required from you. Please note that we will not be able to formally accept your manuscript and schedule it for publication until you have completed any requested changes.

I also want to take this moment to say a quick "Hi" and to (re)introduce myself. I recently started as a PLOS Biology editor, replacing Gabriel Gasque who has moved on to a different adventure. I was, however, Deputy Editor at Neuron under Katja Brose for many years where I recall interacting with you. I’m excited to now be at PLOS Biology, working as the Neuroscience Senior Editor and Section Manager. I look forward to continued interactions with you on future submissions going forward and would be interested in hearing what else is going on in your lab these days when you have some time.

When submitting the final version of your study, please take a minute to log into Editorial Manager at http://www.editorialmanager.com/pbiology/, click the "Update My Information" link at the top of the page, and update your user information to ensure an efficient production process.

PRESS

We frequently collaborate with press offices. If your institution or institutions have a press office, please notify them about your upcoming paper at this point, to enable them to help maximize its impact. If the press office is planning to promote your findings, we would be grateful if they could coordinate with biologypress@plos.org. If you have previously opted in to the early version process, we ask that you notify us immediately of any press plans so that we may opt out on your behalf.

Sincerely, 

Kris

Kris Dickson 

Senior Editor 

PLOS Biology

kdickson@plos.org